# Development of Feline Ileum- and Colon-Derived Organoids and Their Potential Use to Support Feline Coronavirus Infection

**DOI:** 10.3390/cells9092085

**Published:** 2020-09-12

**Authors:** Gergely Tekes, Rosina Ehmann, Steeve Boulant, Megan L. Stanifer

**Affiliations:** 1Institute of Virology, Justus Liebig University Giessen, 35390 Giessen, Germany; 2Bundeswehr Institute of Microbiology, 80937 Munich, Germany; RosinaEhmann@bundeswehr.org; 3Department of Infectious Diseases, Virology, Heidelberg University Hospital, 69120 Heidelberg, Germany; s.boulant@dkfz.de; 4Research Group “Cellular Polarity and Viral Infection”, German Cancer Research Center (DKFZ), 69120 Heidelberg, Germany; 5Department of Infectious Diseases, Molecular Virology, Heidelberg University Hospital, 69120 Heidelberg, Germany

**Keywords:** feline coronavirus, feline enteric coronavirus, FECV, feline infectious peritonitis virus, FIPV, feline intestinal organoids

## Abstract

Feline coronaviruses (FCoVs) infect both wild and domestic cat populations world-wide. FCoVs present as two main biotypes: the mild feline enteric coronavirus (FECV) and the fatal feline infectious peritonitis virus (FIPV). FIPV develops through mutations from FECV during a persistence infection. So far, the molecular mechanism of FECV-persistence and contributing factors for FIPV development may not be studied, since field FECV isolates do not grow in available cell culture models. In this work, we aimed at establishing feline ileum and colon organoids that allow the propagation of field FECVs. We have determined the best methods to isolate, culture and passage feline ileum and colon organoids. Importantly, we have demonstrated using GFP-expressing recombinant field FECV that colon organoids are able to support infection of FECV, which were unable to infect traditional feline cell culture models. These organoids in combination with recombinant FECVs can now open the door to unravel the molecular mechanisms by which FECV can persist in the gut for a longer period of time and how transition to FIPV is achieved.

## 1. Introduction

The diverse family of the *Coronaviridae* causes infections in a wide range of mammals, birds and humans. Feline coronavirus (FCoV) is a highly prevalent member of the *Coronaviridae* family and is found in both domestic and wild cat populations worldwide [1]. FCoVs occur in two different biotypes: feline enteric coronavirus (FECV) and feline infectious peritonitis virus (FIPV). FECV infections commonly manifest as mild or asymptomatic infections of the feline enteric tract. Infections are often persistent and display intermittent shedding of virus over long periods of time which greatly contributes to the high seropositivity levels found in domestic cats. In single household cats, 20–60% of cats display signs of exposure to the virus and up to 90% of cats in multi-cat populations are seropositive [2,3,4].

FIPV emerges through mutations from the harmless FECV and can lead to a fatal clinical condition known as feline infectious peritonitis (FIP) [5,6,7,8,9]. Although the molecular pathogenesis of FIP is poorly understood [10,11,12], promising therapeutical approaches have recently been described [13,14,15,16,17,18]. It is important to note that both biotypes exist in two serotypes [19,20,21]. Serotype II FCoVs are the results of recombination between a serotype I FCoV and a closely related canine coronavirus (CCoV) [22,23,24,25] and can easily be grown in vitro. In sharp contrast, the more relevant and prevalent serotype I FCoVs cannot be propagated in cell culture. Accordingly, serotype II viruses were often used in the past to gain insight into FCoV biology instead of serotype I FCoVs. To elucidate the molecular pathogenesis of FIP, cell culture-adapted serotype I FIPV laboratory strains were obtained over time [26]. However, these viruses proved to be unsuitable to study the pathogenesis of FIP due to the loss of pathogenicity via cell culture adaptation [1,9,26]. The first reverse genetic system that enabled genetic manipulation of the entire FCoV genome was described by Tekes et al. (2008) for serotype I FIPV laboratory strain Black using a vaccinia virus vector [26,27,28]. However, animal experiments showed that like many other laboratory strains, serotype I FIPV Black lost its capability to induce FIP [26]. On the contrary, another commonly used serotype II FIPV laboratory strain, 79-1146, [26,29,30] is much more pathogenic and thus does not appropriately resemble most of the field strains either. Due to the lack of suitable in vitro systems for field serotype I FECVs, it is critical to establish a suitable in vitro system that enables the growth of serotype I FECV. This culture system for serotype I FECV field viruses would not only provide insight into the molecular mechanism by which FECVs persist in the gut for a longer period of time but it might also contribute to the understanding of how FIPV can evolve from FECV during a harmless persistent infection.

Over the past years, organoids have been employed as an in vitro system to support the growth of several human viruses that were unable to be cultured using standard cell culture methods [31,32]. Organoids are derived from either induced pluripotent stem cells (iPSCs) or from tissue-derived stem cells, which are grown and differentiated as three-dimensional structures that closely recapitulate the cellular composition and functions of their originating organ [33]. Tissue-derived organoids rely on the ability to isolate stem cell containing crypts. These crypts are then grown in the presence of differentiation factors (Wnt3a, R-Spondin, Noggin and EGF), allowing them to grow into three-dimensional mini-gut organoids [33]. As these complex cultures more closely resemble the multi-cell types found in their natural tissue counterparts, they often contain factors, which are required for the replication and propagation of viruses that are missing in standard cell cultures. To determine if these model systems could be used to support the relevant serotype I FECV growth, we established a cat intestinal organoid culture system and show that it is capable of supporting infection with GFP-expressing recombinant serotype I FECV generated by reverse genetics. This model will now open the doors to study the molecular mechanism of serotype I FECV-persistence in its natural enteric environment.

## 2. Materials and Methods

### 2.1. Viruses and Cell Lines

Serotype I recFECV-GFP and recFECV-S_79_-GFP were produced in vitro using the reverse genetic system for FCoV field strains described previously [34]. Recombinant virus stocks of recFECV-S_79_-GFP were titrated by plaque assay on routinely used felis catus whole fetus (FCWF) cells [26,27,28,35,36]. Virus stocks of recFECV-GFP which cannot be cultivated in standard cell culture systems were quantified by comparative Western blot analysis of the FCoV M protein together with recFECV-S_79_-GFP [34]. The cell line FCWF was provided by the diagnostic laboratory at the Justus Liebig University Giessen and maintained in culture media (DMEM with 1× penicillin/streptomycin (Thermo, Waltham, MA, USA) and 10% FBS (Biochrom, Cambridge, UK)). According to our experience with propagation of FCoV laboratory strains, the FCWF cells were used at a confluency of approximately 90% for the infection with recFECV-S_79_-GFP.

### 2.2. Animals

Handling of the animals used for enteric tissue donation was performed according to the guidelines of the Hungarian legislation on animal protection. The protocol was approved by the Pest Megyei Kormany-hivatal, Budapest (assurance numbers PE/EA/2441-6/2016 and TMF/657-12/2016). The animals were euthanized according to the designate protocol. Female specific-pathogen-free (SPF) cats were raised and housed in pathogen free conditions for laboratory use. These were not cats taken from outside veterinary practices. The SPF cats were euthanized at an age of 25 weeks and tissue samples from the gut were collected, 6 donors were used for this study. The cats were not tested prior to isolation specifically for FCoV. Animals had access to feed ad libitum prior to euthanization and tissue collection.

### 2.3. Chemicals and Solutions

Conditioned media containing Wnt3a, R-Spondin and Noggin was produced from the L-WRN cell line (ATCC CRL-3276) as per manufacturer’s instructions. A 293T cell line which produces only R-Spondin was a kind gift from Calvin Kuo (Standford University) and conditioned media was made as previously described [37]. All organoid media were made from a base of advanced DMEM/F12 (Thermo, Waltham, MA, USA) which contained 1% penicillin/streptomycin (Thermo), 2 mM GlutaMAX (Thermo) and 10 mM HEPES (Thermo) and is referred to as Ad DMEM/F12++. All other organoid media components are found in Table 1.

### 2.4. Isolation of Feline Intestinal Cells

Ileum and colon sections (10 cm each) were harvested from 6 donors and stored in cold transport buffer (1× phosphate buffered saline (PBS), 50 ng/mL gentamicin (Thermo), 1% penicillin/streptomycin (Thermo), 1% fetal bovine serum (FBS) and 250 µg/mL Fungizone (Thermo)) until the time of isolation. The tissue was cut in half and washed 3 × 10 min with shaking in cold PBS to ensure that all fecal material had been removed. Crypts containing stem cells were isolated within 16 h of sacrificing the animals by first cutting the tissue into smaller 1 cm pieces and then either adding 2 mM EDTA to tissue sample for 1 h at 4 °C or 20 mL of Gentle Cell Dissociation Reagent (Stem Cell Technologies, Vancouver, BC, Canada) for 1 h at room temp. Tissue sections were transferred to a clean tube and 10 mL of cold PBS + 1% BSA was added to tube. Tubes were shaken to release the crypt fraction. Fractions enriched in crypts were filtered with 70 µm filters and the process was repeated to collect four to five fractions for each tissue. The fractions were observed under a light microscope and those containing the highest number of crypts were pooled and spun at 500× *g* for 5 min at 4 °C. The supernatant was removed, and crypts were washed 1× with cold DMEM/F12 (Thermo) and spun at 500× *g* for 5 min at 4 °C. The media was removed, and crypts were re-suspended in 100% Matrigel, plated in 50 µL drops in 24-well non-tissue culture treated plates (Corning) and following polymerization of the Matrigel, 500 µL organoid media was added to each well and was replaced every 48 h. Following isolation of the crypts and seeding into organoid media, the size of the organoids was monitored over time at day 3, 6, 9, 12, and 15 days by observing them under bright field microscopy with a Nikon Eclipse Ti-S microscope. Their size was measured with a 10× objective using the Nikon NIS software.

### 2.5. RNA Isolation, cDNA, and qPCR

RNA was harvested from cells using NucleoSpin RNA extraction kit (Macherey-Nagel, Dueren, Germany) as per manufacturer’s instructions. cDNA was made using iSCRIPT reverse transcriptase (BioRad, Hercules, CA, USA) from 250 ng of total RNA as per manufacturer’s instructions. Quantitative-PCR was performed using iTaq SYBR green (BioRad) as per manufacturer’s instructions, GAPDH was used as normalizing gene. Pre-designed feline specific KiCqStart SYBR Green pre-designed primers were purchased from Sigma-Aldrich (Table 2).

### 2.6. Passaging of Feline Mini-Gut Organoids

Ileum and colon mini-gut organoids were monitored under a light microscope and were passaged when centers became dark and filled with dead cells. For passaging, media was removed and Matrigel was dissolved in cold PBS. Organoids were spun at 500× *g* for 5 min at 4 °C and PBS was removed. Subsequently three different methods were used for passaging:i.Mechanical passaging: 1 mL of cold PBS was used to re-suspend the organoids. Using a 27-gauge needle on a 1 mL syringe, organoids were broken down by passing the solution up and down 10 times through the needle. Organoids were then spun at 500× *g* for 5 min at 4 °C. The supernatant was removed, and the crypts were re-suspended in 100% Matrigel, plated in 50 µL drops in 24-well non-tissue culture treated plates (Corning) and following polymerization of the Matrigel, organoid media 500 µL was added to each well.ii.Trypsin-based passaging: 1 mL of cold PBS was used to re-suspend the organoids. Organoids were then spun at 500× *g* for 5 min at 4 °C. Organoids were washed a second time in cold PBS by resuspending the pellet in 1 mL of PBS and spinning at 500× *g* for 5 min at 4 °C. The supernatant was removed and organoids were incubated in 0.05% Trypsin-EDTA (Gibco) for 5 min at 37 °C. Trypsin digestion was stopped with the addition of serum containing media and samples were spun at 500× *g* for 5 min at 4 °C. Organoids were washed a second time in cold PBS by resuspending the pellet in 1 mL of PBS and spinning at 500× *g* for 5 min at 4 °C. The supernatant was removed, and the crypts were re-suspended in 100% Matrigel, plated in 50 µL drops in 24-well non-tissue culture treated plates (Corning) and following polymerization of the Matrigel, 500 µL organoid media was added to each well.iii.Gentle Cell Dissociation Reagent method: Media was removed and Gentle Cell Dissociation Reagent (Stem cell technologies) was added to the organoid containing pellet and incubated for 10 min at room temp. Organoids were spun at 500× *g* for 5 min at 4 °C and the supernatant was removed. Organoids were washed in DMEM/F12 and then spun at 500× *g* for 5 min at 4 °C. The supernatant was removed and the crypts were re-suspended in 100% Matrigel, plated in 50 µL drops in 24-well non-tissue culture treated plates (Corning) and following polymerization of the Matrigel, 500 µL organoid media was added to each well.

### 2.7. Infection of Cell Culture with Recombinant Viruses

Approximately 90% confluent monolayers of FCWF cells were washed with serum free DMEM culture media. Cells were inoculated with recFECV-GFP and recFECV-S_79_-GFP at a multiplicity of infection (MOI) of 0.01 or serum free culture media for the mock control. The infection was incubated for one hour and subsequently the inoculum was replaced by culture media containing FBS. The formation of plaques and the GFP signal was monitored 48 hours post-infection (hpi).

### 2.8. Infection of Organoids

Organoids were removed from Matrigel by adding cold PBS for 5 min, liquefied Matrigel and organoids were separated by centrifugation (500× *g*, 5 min). Supernatant and Matrigel were removed and organoids were resuspended in media and gently disrupted with a 27 G needle to allow virus to access both the apical and basolateral sides of the organoids. Following disruption, organoid media containing 10^4^ pfu of FCoV was added to the organoids and allowed to incubate for 6 h in suspension. Following the 6-h incubation, Matrigel was added back to the cultures and organoids were observed over a three-day period.

### 2.9. Statistical Methods

Statistics were calculated by Prizm using an unpaired *t*-test.

## 3. Results and Discussion

Currently, infectious disease research of feline intestinal pathogens has been hampered by the lack of cell-based systems allowing for the screening of viruses and antiviral compounds. To fill this gap, we developed a method to isolate and propagate ileum and colon mini-gut organoids from felines. Six SPF cats were euthanized and the ileum and colon were collected. Tissue sections were washed thoroughly to remove all fecal material and were stored at 4 °C in a transport buffer containing PBS as well as antibiotics and antifungals. Organoids were prepared within 16 h of tissue harvesting. To determine the best method to isolate crypts containing stem cells from feline intestinal tissues, samples were split in two parts and two commonly used approaches (EDTA-based dissociation and Gentle Cell Dissociation Reagent™) for harvesting human and murine intestinal crypts [38,39] were compared side by side. Feline ileum and colon samples were washed thoroughly to remove all mucus or remaining contaminants and crypts containing stem cells were then isolated following incubation in EDTA or Gentle Cell Dissociation Reagent™. Microscopic evaluation of the crypts prior to seeding showed no large difference in the quantity or quality of crypts between the two isolation methods (data not shown). Equal numbers of crypts were seeded into Matrigel and the number and size of organoids was followed over 12 days. Unlike human and murine crypts, which form organoids with 16–24 h [38,39,40], the feline crypts took 72 h to seal and form cyst like organoids (Figure 1A). Following initial cyst formation, the ileum and colon organoids continued to grow over the twelve-day period (Figure 1A). To determine which isolation method produced the best organoids, the number and size of organoids was followed over time. Results show that organoids isolated using the EDTA approach produced a greater number of organoids and the organoids were larger in size compared to the Gentle Cell Dissociation Reagent™ method (Figure 1B,C). The increase in the number of organoids between day 3 and 6 does not illustrate a growth in organoid number but is due to the fact that at day 3 the crypts are too small to be effectively counted. This number of organoids then remained constant over the rest of the period as they continued to grow larger (Figure 1B,C), which was consistent with murine and human organoid generation [38,39].

To determine if the organoids displayed similar cell types as the natural feline intestine, tissue and organoids samples from three donors were lysed and the relative expression of each cell type was quantified by q-RT-PCR. As the natural intestine is made of stem cells, absorptive enterocytes and secretory cells we evaluated known markers from each of these populations. As there are no markers to analyze feline organoids, we used the best-known markers for murine and human, hypothesizing that they will constitute a marker for feline tissue. As a marker of stem cells, we used LGR5 and SMOC2. LGR5 is the historical marker described by the Clevers lab, however while it constitutes an excellent marker in tissue it is known to be suboptimal for organoids [38,39,41]. On the contrary, SMOC2 represents an excellent marker for both tissue and tissue-derived organoids. As a marker for Goblet we used mucin 2 (Muc2), for enterocytes we used sucrose isomaltase (SI), for enteroendocrine cells we used synaptophysin (SYP) and for Paneth cells we used lysozyme (LYZ). Results showed that ileum from feline tissue samples displays markers for stem cells (LGR5, SMOC2), Goblet cells (Muc2), enterocytes (SI), enteroendocrine cells (SYP) and Paneth cells (LYZ) (Figure 2A). Similarly, feline colon tissues expressed similar amounts of markers for all cell types except Paneth cells which are not present in colon tissue. The ratios found in feline tissues are similar to those observed in humans [41].

Importantly, organoids derived from feline ileum expressed the same cell type-specific markers as their tissue counterpart (Figure 2B). The relative expressions of the different markers were slightly different compared to the expression in tissue (Figure 2A,B). Ileum organoids displayed a higher relative expression of the stem cells marker (SMOC2) and a lower relative expression of the enterocyte maker (SI) compared to normal feline tissue. This suggests that there are more stem cells than enterocytes. The increase in stem cell number is expected as organoids are cultured under high Wnt3a conditions which favor stem cell numbers. Similarly, colon organoids also displayed a high amount of stem cell markers and their cellular composition looked similar to those of ileum organoids (Figure 2B). These differences in expression of the different cell type specific markers are not specific to the feline ileum and colon organoids but this is also observed in both murine and human intestinal organoids [40,42]. This shows that although organoids are extremely close to the originating tissue, the fact that they are ex vivo mini-organs causes subtle differences in their expression profile and differentiation pathways. All together these data show that we have developed a novel protocol to isolate and generate intestinal organoids from feline intestinal tissue and that these organoids closely resemble their tissue counterpart.

### 3.1. Passaging and Maintenance of Feline Intestinal or Ganoids

As the ileum and colon organoids continued to grow and retain a similar identity to the natural tissue, we wanted to determine whether they could be passaged and maintained in culture. Trypsinization, mechanical disruption and passaging using Gentle Cell Dissociation Reagent™ were tested to establish the method that best supported the continued maintenance of feline ileal and colon organoids. For all methods, organoids were removed from the Matrigel and either incubated with low concentrations of trypsin, mechanical disrupted using a 27 G needle, or were incubated with Gentle Cell Dissociation Reagent™ (see materials and methods for full details). Following disruption, organoids were centrifuged to separate out organoid structures from dead cells. The organoids were then seeded into Matrigel and the formation of new organoids was followed for five days. One day post-passaging many small organoids could be seen in all conditions for both the ileum and colon organoids (Figure 3A). Organoids which were passaged by trypsinization were small and stressed, as shown by their loss of tight borders (dark rim at the periphery of the organoids under phase microscope) and the presence of many dissociated cells. On the contrary, organoids that were obtained using the mechanical disruption and Gentle Cell Dissociation Reagent™ methods were larger and their borders look more discrete (Figure 3A). Similar to the original isolation, many organoids were too small to be counted on the first day after passaging for all methods (Figure 3B). Observations five days post passaging showed that using mechanical disruption for passaging both ileum and colon organoids leads to a greater number of organoids that are larger in comparison to organoids that were passaged using trypsin (Figure 3B,C).

Upon isolation of feline ileum and colon crypts, organoids were maintained in a media based upon one that is commonly used to support the growth of human organoids [39,43,44]. To determine if additional media compositions would support or enhance feline intestinal organoid growth, we tested three different formulations normally used to support human intestinal and colon organoids, murine small intestine or murine colon (see material and methods) [37,38,39,40,45]. Feline ileum and colon organoids were passaged using mechanical disruption and were maintained in each of the media conditions. Ten days post passaging, the number of organoids and the size of the organoids were counted for each condition. Results show that the human media condition supported a greater number and a larger size of both ileum and colon organoids (Table 3). Additionally, to determine how many passages each media type could support, organoids were split using mechanical dissociation and followed over time. The mouse intestine media did not support long term growth of either ileum or colon organoids and both types of organoids stopped growing and died within two passages in this media (Table 3). The mouse colon media supported longer passaging than its intestine counterpart, however this was still limited to a few passages. The human media was the only one that supported longer term passaging. However, unlike human and mouse organoids [38,39,40], both the feline ileum and colon organoids became arrested after 14–15 passages (Table 3). This was reproduced with several animals (n = 6) and was consistent between animals suggesting that an additional media component may be required to support culturing to the extent that human and murine organoids can reach. Importantly, it seems that Wnt3A is a critical component required for the maintenance of feline organoids as the mouse intestine media, which lacks Wnt3A, supported the fewest number of passages. Overall, we could see that feline intestinal and colon organoids could be maintained using mechanical disruption and Wnt3A containing media. In recent years, intestine- and colon-derived organoids have been generated from both large farm animals (bovine and porcine) [46] and domestic animals (canine) [47,48]. Bovine, porcine and canine intestinal and colon organoids have been shown to require Wnt3a for their generation and growth [46,47,48]. Additionally, while many of these studies did not compare different passaging techniques, they often used mechanical disruption as a common method to passage and maintain the organoids [46,47].

### 3.2. Infection of Feline Intestinal Organoids

Currently, the molecular mechanism by which FECVs can persist in the gut could not been studied due to the lack of cell culture models supporting the replication of FECV field viruses. To evaluate if the feline intestinal and colon organoid system could support FECV infection, two different FCoV viruses (recFECV-GFP and recFECV-S79-GFP) were constructed for this study (Figure 4A). recFECV-GFP is a recombinant FCoV with the entire genome sequence of a serotype FECV field strain while recFECV-S79-GFP contains the genomic backbone of the same serotype I FECV field strains with the spike protein gene of cell-culture-adapted serotype II laboratory strain 79–1146. To determine the ability of recFECV-GFP and recFECV-S79-GFP to replicate in cell culture, these recombinant viruses were first tested on FCWF cells. Inoculation of FCWF cells with recFECV-GFP, did not lead to any formation of cytopathic effect (CPE) or fluorescence as standard cell culture systems do not support the growth of serotype I FCoV field strains (Figure 4B), while as expected, recFECV-S79-GFP growth was supported in FCWF cells demonstrated by their ability to form a characteristic plaque phenotype accompanied by green fluorescence (Figure 4B).

Previously, Desmarets et al. (2013) [49] established permanent feline intestinal epithelial cell cultures of ileocyte and colonocyte origin that were shown to sustain propagation of FECV field strains in vitro. However, such valuable cell lines are not broadly available to the scientific community. To test if the established feline ileum and colon organoids could be used to support feline coronavirus infection, five days post-passaging, organoids were removed from the Matrigel and incubated with feline coronavirus recFECV-GFP and recFECV-S79-GFP and followed over 72 h. Unexpectedly, no fluorescence was obtained upon infection of ileum organoids with either recFECV-GFP or recFECV-S79-GFP (Figure 4D). The lack of fluorescence of ileum organoids upon infection with both viruses is somewhat surprising, since ileum has been described as a tissue supporting FCoV-growth in vivo [50]. Whether the lack of green fluorescence of ileum organoids is due to (i) a very low viral replication rate and GFP expression which is below the detection limit or (ii) the complete lack of virus infection, cannot be ruled out in the current experimental setup. For example, luciferase-expressing recFECVs might be more suitable tools to further investigate these hypotheses. The low/no infection of ileum organoids with the recombinant viruses may reflect the limitation of ileum organoids to study the biology of serotype I FECVs in vitro. In contrast, both recombinant viruses were found to infect feline colon organoids (Figure 4C,D). One day post-infection, GFP positive cells could be detected and the number of infected cells increased over three days for both viruses (Figure 4C,D). As opposed to the FCWF cells, our feline colon organoids supported infection of recFECV-GFP. Since the number of GFP positive cells increased over time, this data strongly suggests that colon organoids support initial infection, production of de novo virus and spreading of serotype I FECV field strain-infection.

In summary, we propose that feline colon-derived organoids represent a primary intestinal cell model supporting serotype I field stain FECV infection. These cultures now open the doors for researchers to unravel the molecular mechanisms leading to FECV persistence and possibly FECV-FIPV conversion. The here described protocol provides the community with a step-by-step approach to generate feline colon-derived organoids which provides a solution for the lack of easily available culture models. Furthermore, as primary intestinal epithelium cells are often more immune-responsive than their immortalize counterparts, as such, the here described organoid model will allow the community to better study host/pathogen interactions [37] and immune response [40,51] against FCoV.

## Figures and Tables

**Figure 1 cells-09-02085-f001:**
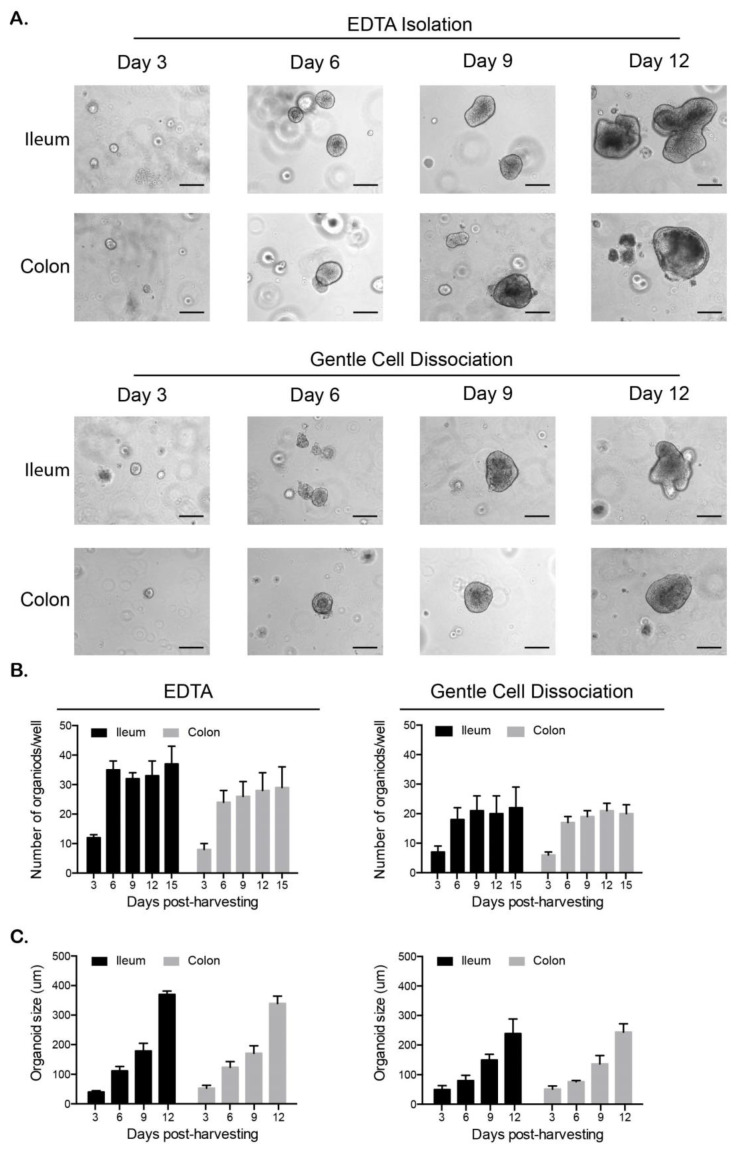
Ileum and colon organoids can be established from primary feline tissue. (**A**–**C**). The 1 cm^2^ sections of feline ileum and colon tissue were incubated with EDTA or Gentle Cell Dissociation Reagent to allow for the isolation of intestinal crypts. Isolated crypts were resuspended in Matrigel and followed over a 12-day period. (**A**). Bright field images of ileum and colon organoids. Representative images are shown (n = 6 donors). Scale bar = 100 µm. (**B**). The number of organoids from EDTA and Gentle cell Dissociation Reagent isolation were counted over the indicated time course. Error bars represent standard deviation (n = 6 donors). (**C**). Following EDTA and Gentle cell Dissociation Reagent isolation, organoids were imaged in three-day intervals using a Nikon Eclipse Ti-S. Their size was measured using the Nikon NIS software. Error bars represent standard deviation (n = 6 donors).

**Figure 2 cells-09-02085-f002:**
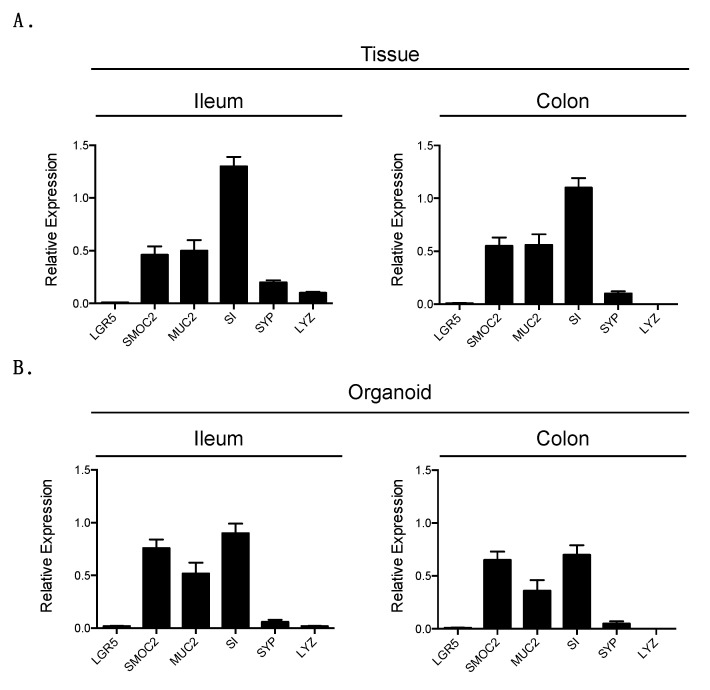
Feline organoids and originating tissue contain similar cell types. (**A**). The relative expression of intestinal cell type specific markers was evaluated in feline ileum and colon tissue by q-RT-PCR. (**B**). Same as A except using feline ileum- and colon-derived organoids. Results are mean +/− s.d. and are expressed as a relative expression to the house-keeping gene GAPDH. n = 3 donors and q-RT-PCR was done as technical triplicate.

**Figure 3 cells-09-02085-f003:**
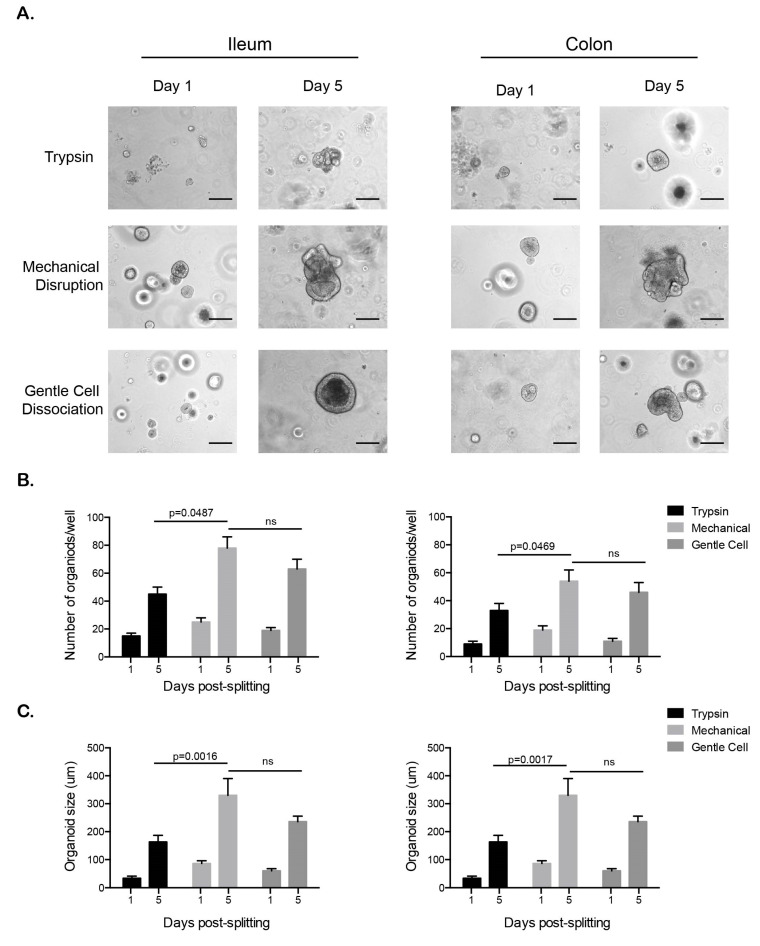
Mechanical disruption is the preferred method for passaging feline organoids. (**A**–**C**). Feline ileum- and colon-derived organoids were grown for 10 days post-harvesting prior to passaging. Three methods of passaging were used, and new organoids were followed over five days. (**A**). Bright field images of ileum and colon organoids. Representative images are shown. N = 3 donors and each donor were followed overtime as a triplicate series. Scale bar = 100 µm (**B**). The number of organoids from each passing method was counted over the indicated time course. Results are mean +/− s.d. (n = 3 donors). (**C**). The size of the organoids was determined using a Nikon Eclipse Ti-S. Results are mean +/− s.d. (n = 3 donors).

**Figure 4 cells-09-02085-f004:**
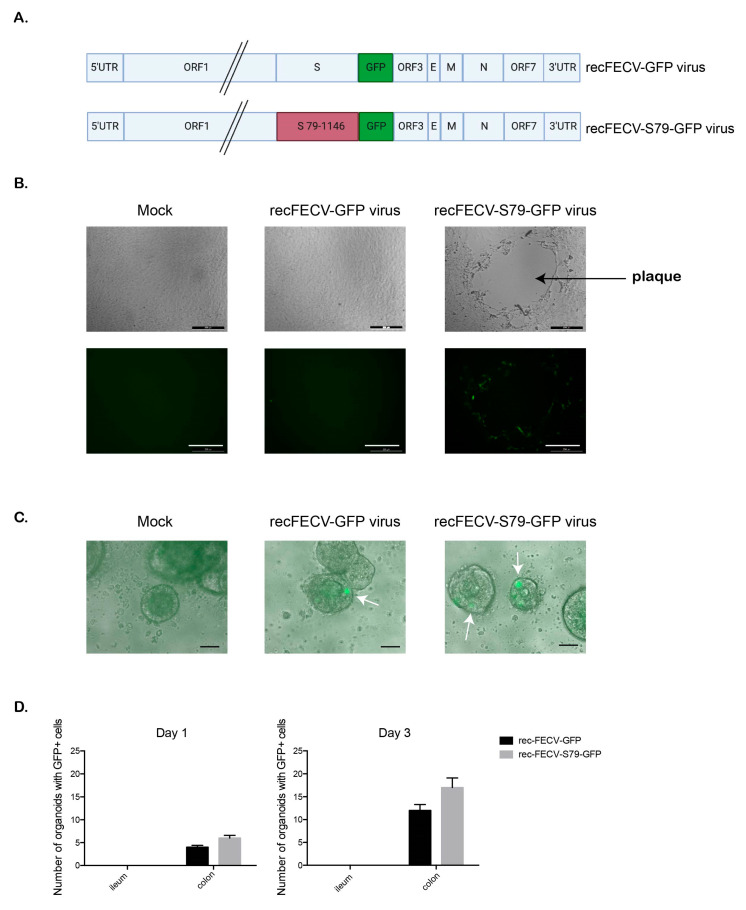
Feline colon organoids support infection of feline coronavirus (FCOV). (**A**). Diagram depicting the newly generated feline enteric coronavirus (FECV) recombinant viruses. (**B**). Felis catus whole fetus (FCWF) feline cells were infected with rec-FECV-GFP or recFECV-S79-GFP viruses and their ability to express GFP and produce plaques was monitored 48 h post-infection. Mock samples indicate media alone conditions. Representative images are shown. Scale bar is 200 µm. (**C**). Feline colon organoids infected with rec-FECV-GFP or recFECV-S79-GFP for 72 h. Representative images shown. White arrow heads indicate GFP positive cells. Scale bar = 100 µm (**D**). Feline ileum and colon organoids (passage 3) were infected with rec-FECV-GFP or recFECV-S79-GFP viruses. Mock samples indicate media alone conditions. Infections were monitored 1 and 3 days post-infection by wide field microscopy. Results are mean +/− s.d. n = 3 donors for each infection.

**Table 1 cells-09-02085-t001:** Human and mouse media compositions tested for their ability to support feline ileum and colon organoid growth. Final concentrations and manufactures of each component are listed.

Reagent	Company	Final Concentration	Human Media	Mouse Intestine Media	Mouse Colon Media
L-WRN (Wnt3s, R-Spondin, Noggin containing conditioned media)	Made in the lab	50%	X		X
Ad DMEM/F12++	Thermo	50% (Human and mouse colon) 90% for mouse intestine	X	X	X
B27	Thermo	1X	X	X	X
Nicotinamide	Sigma (Munich, Germany)	10 mM	X		
N-acetylcysteine	Sigma	1 mM	X	X	X
A-83-01	Tocris (Bristol, UK)	500 nM	X		
SB202190	Sigma	500 nM	X		
Leu-Gastrin	Sigma	10 nM	X		
Mouse recombinant	Thermo	50 ng/mL	X	X	X
R-Spondin conditioned media	Made in the lab	10%		X	
Mouse recombinant Noggin	Peprotech (Rocky Hill, NJ, USA)	100 ng/mL		X	
Y-27632	Sigma	10 µM	X	X	X
Matrigel, Growth factor reduced (GFR), phenol free	Corning (Corning, NY, USA)	100%	X	X	X

**Table 2 cells-09-02085-t002:** Feline specific primers used to control cell population in tissue and organoids.

Gene	Cell Type	Gene ID	Sequence ID
GAPDH	House keeping	493876	NM_001009307
LGR5	Stem cell	101080720	XM_003989046
SMOC2	Stem cell	101082409	XM_003986725
MUC2	Goblet cell	101096605	XM_003993797
SI	Enterocyte	100144605	NM_001123332
SYP	Enteroendocrine	101084343	XM_004000526
LYZ	Paneth cell	100127109	XM_003989032

**Table 3 cells-09-02085-t003:** Optimization of feline ileum and colon culturing conditions. The number and size of organoids were measured 10 days post-splitting for every other passage.

	Organoid	Avg # of Organoids/Well	Avg. Size of Organoids	No of Passages
Human Media	Colon	83 +/− 11	384.2 +/− 31.5	15
Mouse Intestine Media	Colon	54 +/− 6	312.5 +/− 23.6	2
Mouse Colon Media	Colon	68 +/− 9	359.4 +/− 29.1	6
Human Media	Ileum	68 +/− 9	346.2 +/− 24.8	14
Mouse Intestine Media	Ileum	49 +/− 6	297.6 +/− 21.3	2
Mouse Colon Media	Ileum	53 +/− 7	318.9 +/− 35.4	4

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
