# Peer review of "Development of Feline Ileum- and Colon-Derived Organoids and Their Potential Use to Support Feline Coronavirus Infection"

_cells, 2020, doi:10.3390/cells9092085_

Round 1

Reviewer 1 Report

The manuscript “Development of feline ileum- and colon-derived organoids to support feline coronavirus infection” describes how the authors have achieved in vitro stable cellular structures in which feline coronaviruses may replicate. This may be of special consequences to study the pathogenicity of these viruses, as they mutate from the normal enteric biotype into the more pathogenic variety which produces feline infectious peritonitis, and test potential antivirals. The results are interesting though the manuscript itself lacks substantial information and needs to be reviewed by the authors in order to complete necessary data.

Major concerns are:

  • It is not unmistakably proven that the animals where the organoids came from were FCoV- free. Please, provide data.
  • The viruses used to test the ability of organoids to support their replication are laboratory constructs and not field strains. In fact, one of the two constructs is only partly a feline coronavirus. Thus, the experiment was only done with one virus, which is not representative. As such, the title should be changed to “Development of feline ileum- and colon-derived organoids and their potential use to support feline coronavirus infection”
  • The introduction lacks information about organoids, what are they, how are they usually achieved.
  • The manuscript mistakes continuously crypts, stem cells and organoids. As I understand it, crypts are a defined part of the intestinal mucosa, containing enteroblasts, which may also be called stem cells, which generate enterocytes that migrate to the villi of the folds of the mucosa. It is incorrect that the authors say “the fractions with the highest number of crypts”. These should be reviewed throughout the manuscript and corrected.
  • Material and methods are very poorly described.
  • The present tense used for results is not adequate.
  • The discussion with the results of other authors is virtually non-existent.

Specifically, (when copying from the word do

L18. could not be → may not be

L19-21. The sentence is not relevant for the abstract

L27. and transition to FIPV → and how transition to FIPV is achieved

L38. high seropositivity levels → high seropositivity levels found in domestic cats.

L38. Rephrase so that it does not start with a number.

L42-43. Please, remove Importantly and change are not able to by cannot.

L44. Please, provide reference.

L46. Please, provide reference.

L47. has been → was

L51. Please, remove Importantly.

L52. remain → remains

L51-53. The sentence is misleading as the rest of the work does not support this statement.

L53. critical that we → critical to

L58-65. Please, provide more information about organoids, what they are, how they are obtained, etc.

L63. Please, remove of.

L69. Please, describe fcwf the first time that it appears. It would be helpful if all the letters are capitalized. Indicate how do they grow, type of passage (1:3, 1:5?) and how long they take to reach confluency, along with a reference.

L75. Please, clarify why the Hungarian legislation on animal protection was followed instead of the German one.

L77. Please, provide more information about the SPF cats, how old they were, their sex, and how they were euthanized. Very important, describe the tests done to prove that they were not infected by FECV. Indicate whether or not they had fasted.

L79-81. It is not clear which cells produce R-Spondin, Noggin and which were a gift, who is the manufacturer. Please, rephrase. If a manufacturer is involved, please, cite. Complete the affiliation of Dr.? Calvin Kuo.

L82. medias → media.

L85-86. Please, label the table and indicate abbreviations. It would be helpful if the media for organoids was named in a specific way. Were all the reagents added to the medium or only some of them, leaving others for other uses? Where was Matrigel diluted in?

L86-99. This is the most important part of the paper. Please, describe in full, including how the sections were thoroughly washed to eliminate fecal material, the size of the sections, etc.

L89. Change to Crypts containing stem cells

L91. moved → transferred.

L92. Add s to all tube → tubes.

L93, 95, 96, 97, 115, 125, 131.. crypts → stem cells

L96. Be consistent with the name of the medium.

L98. Please, indicate the percentage of Matrigel used at each step throughout Materials and Methods.

L98-99. Is organoid media the same as culture medium? Please, use a consistent term. Please, describe the criteria for considering a clump of cells an organoid. Indicate too, how many cells per well and well size (size of plate).

Please, indicate how were organoids measured.

L100-105. Please reorder because in the present order it is not q-RT-PCR.

Table 1. In its present state, the only useful columns of this table are the first two. It does not describe the primers used, indicate the source, the sequence, the size of the amplicon, may be even the catalog number. Please, indicate also if all the genes are feline or some are human.

Please, add statistical methods.

L107. As the intestine is a very broad term, please, change to ileum.

L112, 125, 131. Indicate in all three procedures “re-suspended in Matrigel, placed in wells (size of plate) and following…”

L128. Please, clarify stem cell.

L140. Please, indicate which medium.

L149. Six SPF cats…collected.

L149. Please, always specify if “small intestine” is just “ileum” or also other parts.

L154. Provide references after murine intestinal cells.

L155 and thereafter. When naming the Gentle Cell Dissociation always complete it with either Reagent or Media, as otherwise it suggests that it is a procedure.

L152-157. Please, rephrase. Also L157-159 to indicate that it is stem cells which are being evaluated. And provide data comparing ileum and colon.

L159. Please, indicate how many cells.

L159. Please, change crypts → stem cells

L161. Please, provide reference (and remove Interestingly,)

L165-166. Please, rephrase.

L170. Please, compare to other authors and their results.

Figure 1. A) Indicate a bar for size. A) and B) Modify Gentle Cell Dissociation. B) and C) (both can be B)) Change in the y axis to Number. D) and E) can be one. Thus, it would be only A, B, and C.

L199. Please, discuss and explain what is the difference between markers SMOC2 and LGR5. Discuss whether markers used are specific for feline cells. Three donors do not seem enough for this type of study. Indicate why not all six donors were tested. Discuss distribution of cell populations with more authors (Angus et al., 2020; Meneses et al., 2016).

L201. Explain what is Wnt3a.

L229. Figure 3B and 3C.

Figure 3. The same comments as for Figure 1. Please, include statistical data.

L242. Please, provide references just before (Table 2).

L246, 250, 253. Table → Tables

L248. Please, remove Interestingly,

L252. passaging (Table 2). However, unlike human and mouse organoids (references).

L259. Please, provide more discussion with other authors and maybe other animal species or origin of stem cells for organoids.

Table 2. This table should be in Materials and Methods, combined with the unnumbered Table. In any case, abbreviations should be explained and the connection with Wnt3A.

L263, L265. Please, indicate which passage it is. Combine both tables into a single one.

L271-273. Please, provide references.

Figure 4. Same comments as Figure 1 regarding bar size. Was a statistical analysis performed? GFP signal is very faint. Please, add control (mock inoculated). L284. Indicate the age of the organoids. L287. What is B-D?

L290. Please, be consistent with abbreviations.

L299. Please, provide data as to how the GFP signal increased over time.

L300. that organoids → that colon organoids. Please, discuss in view of the pathogenesis of feline coronaviruses as to why replication may be higher in the colon than in the ileum.

L306. lack easily → lack of easily

L307-309. Please, review sentence.

References 16 and thereafter. Why are co-authors only named as et al.?

I do not seem to be able to find the Supplementary Files.

Author Response

Please see attached pdf file for point by point answers.

Reviewer 2 Report

Manuscript ID: Cells-916734

Communication: Development of feline ileum- and colon-derived 2 organoids to support feline coronavirus infection

General comments:  The present study describes the methodology to establish organoids from feline ileum and colon. Various technical aspects of the generation and maintenance of organoids were compared. Molecular clones of serotype I and serotype II FCoV were then shown to infect the colonic but not the ileum organoids. Overall the manuscript provide a useful method to develop feline intestinal organoids and also provides preliminary data that the colonic organoids might be useful to study the intractable serotype I FECV. In this regard the manuscript contributes new knowledge to the field should be of interest to feline intestinal virus researchers. The manuscript requires a major revision to correct inconsistencies, provide additional detail to enhance understanding, and improve the editing. The authors should pay close detail to the required formatting, abbreviations, units, and supplier references indicated in the journal instructions to authors.  

Specific comments:

Line 41: There are now a small number of reports showing efficacy of two antiviral compounds.

Line 42: It differences between serotype I and serotype II FECV  with regard to in vitro propagation should be describe and it should be clarified that growth of serotype 1 viruses in enteroids is the focus of this report. The serotype of viruses mentioned in the introduction should be identified as serotype I or II.

Line 65: Replace FEVP with FECV.

Line 67: Indicate whether the molecular clones are serotype I or II.

Line 69: Define fcwf in this sentence.

Line 82: Medium is singular, media is plural; please correct throughout.

Table 1: Indicate that the primers were feline sequence specific.

Line 133: In Line 69 it is stated that recFECV-GFP cannot be cultivated in standard cell culture systems but in this section it is stated that virus was titered in fcwf, please clarify how MOI of recFECV-GFP was determined.

Line 151: antifungals and antimycotics are the same thing.

Line 199: The wording suggest cell were quantified based on expression of differentiation transcripts, but this does not seem to be the case. Please reword to accurately reflect results as relative transcript frequency.

Line 301: Please indicate whether infectious virus was produced in organoid cultures.

Discussion on why ileum was not infected should be provided given this has been previously reported.

Author Response

Please see attached pdf document for point by point answers to reviewer comments.

Round 2

Reviewer 1 Report

The manuscript has greatly improved as a result of an obvious effort of the authors. I have re-read it again and find it very interesting. I thank the authors for clarifying about the crypts. I had thought that through the disgregation process, cells would have individualized and only stem cells would continue growing (similar to hybridomas used for monoclonal antibodies). I understand that they grow as a clump and remain that way throughout the process. I suggest that the concept of crypt is clarified in the text. After re-reading, I have some very minor observations.

The use of FCWF, verus fcwf, is a very adequate change and has greatly improved clarity.

L117. I thank the authors for the explanation in private about the cats used and I feel that it should be made public. Please, include the first information provided in the response letter (about the spf cats not being tested specifically for FCoV) in the manuscript.

L121. Change were for was.

L125-129. I suggest combining both tables and labeling as Table 1.

Review the manuscript to change uL to μL

L215, 219. Change crypt for crypts.

L240. Rephrase are no marker. Either are no markers or is no marker.

L251. I think it should be humans.

Table 1 (please, table headings above the table). It seems to be missplaced and to belong to Materials and Methods.

Tables 2 and 3. I still think that they would be better together, with an obvious separation between ileum and colon. 

Author Response

Please see attached document for point-by-point answers to reviewer comments.
